# Ocular Toxoplasmosis Associated with Unilateral Pigmentary Retinopathy That May Mimic Retinitis Pigmentosa: Diagnostic Dilemmas

**DOI:** 10.3390/medicina57090892

**Published:** 2021-08-27

**Authors:** Izabella Karska-Basta, Bożena Romanowska-Dixon, Dorota Pojda-Wilczek, Natalia Mackiewicz

**Affiliations:** 1Department of Ophthalmology, Faculty of Medicine, Clinic of Ophthalmology and Ocular Oncology, Jagiellonian University Medical College, 31-501 Kraków, Poland; romanowskadixonbozena1@gmail.com (B.R.-D.); mackiewicz.nat@gmail.com (N.M.); 2Department of Ophthalmology, School of Medicine, Medical University of Silesia in Katowice, 40-752 Katowice, Poland; pojda-wilczek@wp.pl

**Keywords:** ocular toxoplasmosis, retinitis pigmentosa, unilateral pigmentary retinopathy, unilateral retinitis pigmentosa

## Abstract

We report a unique case of coexisting pigmentary retinopathy and ocular toxoplasmosis in a young male patient. A 23-year-old man presented with sudden visual deterioration in the left eye (LE). The fundus findings revealed bone spicule-shaped pigment deposits, a slightly pale optic disc, arteriole constriction, cystoid macular edema with an epiretinal membrane, and two small inflammatory chorioretinal scars in the right eye, with a concentric narrowing of the visual field and a nonrecordable multifocal electroretinogram (ERG). An active inflammatory lesion at the border of a pre-existing chorioretinal scar in the macula was found in the LE, with a central scotoma in the visual field. Moreover, the patient tested positive for anti-*Toxoplasma gondii* immunoglobulin G antibodies and showed positive results in polymerase chain reaction testing of aqueous humor. Fluorescein angiography revealed hyperfluorescence in the early phase with fluorescein leakage. A multifocal ERG of the LE showed selective loss of responses from the central 10 degrees. Genetic testing revealed heterozygosity in the *RP1* and *CELSR1* genes. Our case illustrates challenges in the diagnosis of unilateral pigmentary retinopathy. Based on the typical toxoplasmic lesions in the LE and two scars likely caused by inflammation, our patient was diagnosed with pigmentary retinopathy probably related to toxoplasmosis. Genetic consultation did not confirm the diagnosis of retinitis pigmentosa, but more advanced tests might be needed to definitively exclude it.

## 1. Introduction

Ocular toxoplasmosis most commonly affects young and previously healthy patients. The typical findings in the active stage of the disease, such as a yellow-white focus with fluffy borders, can also be accompanied by vitritis, which usually limits the visualization of the posterior pole. After the active stage lasting approximately six weeks, pigmented chorioretinal atrophic scars can be found. Ocular toxoplasmosis may present with rare clinical manifestations, such as neuroretinitis, occlusive retinal vasculitis, retinal and subretinal neovascularization, multifocal retinochoroiditis, rhegmatogenous and serous retinal detachment, and pigmentary retinopathy mimicking retinitis pigmentosa (RP) [1,2]. The final diagnosis should be supported by specific serologic testing for anti-*Toxoplasma gondii* antibodies, and in some cases, it should be confirmed by genetic testing of the aqueous humor or vitreous [3].

The term “pigmentary retinopathy” refers to the migration and proliferation of retinal pigment epithelial cells or macrophages containing melanin pigment into the retina. It is mostly observed in various dystrophic, infectious, or other systemic diseases. The presence of retinal dystrophic and pigmentary changes, as well as the frequent association with night blindness, a loss of visual acuity, the constriction of visual fields, and abnormalities on electroretinography (ERG), make pigmentary retinopathy strikingly similar to RP [4]. On the other hand, RP refers to the most common retinal dystrophy, affecting one in 3000 to 7000 individuals and manifesting with abnormalities of the photoreceptors leading to a decrease of night vision, impairment of bilateral visual function, and a gradual concentric loss of peripheral vision. The disorder usually occurs in the first three decades of life. It can be inherited in an autosomal dominant, autosomal recessive, or X-linked manner [5]. The bone spicule pigmentary changes, optic disc pallor, vasoconstriction, macular edema, and subcapsular cataract are typical clinical symptoms found on fundoscopy [6].

## 2. Case Presentation

A 23-year-old male patient presented with sudden visual deterioration in the left eye (LE). His medical and ocular history was unremarkable, and no family history of acquired or inherited diseases was reported. The best corrected visual acuity of the right eye (RE) and LE was 1.0 (20/20) and 0.125 (20/160), respectively. The anterior segment of the right and left eye was unremarkable with no subcapsular cataract. The fundus findings revealed bone spicule-shaped pigment deposits, a slightly pale optic disc, arteriole constriction but mostly in the nasal part of the fundus, cystoid macular edema with an epiretinal membrane, and two chorioretinal scars in the periphery of the RE (Figure 1A and Figure 2A), confirmed by fundus autofluorescence (Figure 1B), with a concentric narrowing of the visual field (Figure 1D) and a nonrecordable multifocal electroretinogram. Active inflammatory lesions (Figure 1A and Figure 2A) at the border of a pre-existing chorioretinal scar in the macula were found in the LE (Figure 2A) (without any abnormalities typical for RP), with a central scotoma in the visual field (Figure 1D). Fluorescein angiography (Figure 1C) showed patchy hypofluorescence due to bone spicule-shaped pigment deposits in the periphery of the RE. It also revealed hypofluorescence in the central part of the macula with hyperfluorescent satellite spots with fluoresceine leakage in the late phase due to active inflammation in the LE (Figure 1C). The swept-source optical coherence tomography showed cystoid macular edema with an epiretinal membrane, a normal retinal pigment epithelium (RPE) and foveal photoreceptors but without a photoreceptor-pigment epithelium complex in the rest of the retina, normal choroidal thickness, and single hyperreflective satellite spots in the RE (Figure 2B). Furthermore, it revealed atrophy of the outer retinal layers, with a hyperreflective area corresponding to pigment deposits in the fovea, and disorganization of the retinal structure with a markedly thickened choroid corresponding to an active inflammatory lesion with hyperreflective satellite spots in the vitreous of the LE (Figure 2B). A multifocal ERG examination was performed according to the International Society for Clinical Electrophysiology of Vision standard [7], using VERIS Clinic 5.0 (Electro-Diagnostic Imaging, Inc.; Redwood City, CA, USA). Corneal ERG-Jet electrodes and binocular stimulation were applied. A multifocal ERG of the RE was nonrecordable, while the ERG of the LE showed selective loss of responses only from the central 10 degrees. It corresponded with the central macular lesions. Moreover, the patient tested positive for serum anti-*Toxoplasma gondii* immunoglobulin G antibodies. Additionally, he showed positive results in polymerase chain reaction testing of aqueous humor samples. However, genetic testing revealed heterozygosity in the *RP1* and *CELSR1* genes. The mother, father, and both sisters of the patient did not develop symptoms of RP. As healthy family members presented with *RP1* and *CELSR1* gene variants (c.742C > T and c.8000C > A, respectively), their pathogenicity should be excluded. Moreover, the whole exome sequencing did not allow us to establish the genetic basis of pigmentary retinopathy in the RE.

## 3. Discussion

Apart from the typical clinical appearance of ocular toxoplasmosis, such as retinochoroiditis with accompanying chorioretinal scaring and a range of vitreous symptoms, numerous so-called atypical presentations of the disease have been reported [3]. Therefore, the patient’s medical history and fundus findings have to be carefully examined and complemented by laboratory testing.

We described a case of ocular toxoplasmosis with a typical clinical picture in the LE, as well as pigmentary retinopathy accompanied by two small inflammatory scars. Already since presentation, the fundus examination findings of the RE, such as bone spicule-shaped pigment deposits, a slightly pale optic disc, arteriole constriction, and cystoid macular edema, strongly resembled unilateral RP. Moreover, the concentric narrowing of the visual field and completely absent multifocal ERG seemed to confirm the diagnosis. As part of further diagnostic workup, genetic tests and consultation were performed. However, they did not support the diagnosis of unilateral RP. It is known that this rare condition can occur only in a single eye, with the other eye remaining unaffected. The strict diagnostic criteria are as follows: (1) functional changes and ophthalmoscopic appearance typical for RP should be present in the affected eye; (2) the absence of RP symptoms in the other eye, with simultaneous normal ERG findings; (3) a sufficiently long follow-up (>5 years) to exclude a delayed onset in the unaffected eye; and (4) an underlying inflammatory cause in the affected eye has to be excluded [8,9]. The last criterion was not fulfilled in our case because there were two small scars in the RE, likely due to inflammation. Moreover, it seems that the presence of cystoid macular edema with an epiretinal membrane, chronic macular edema, and hyperreflective satellite spots in the vitreous suggesting vitritis in the right eye, as well as the absence of typical RP findings, such as a subcapsular posterior cataract or optic disc pallor, may suggest atypical ocular toxoplasmosis that was previously described in the literature. Except for the common clinical signs of ocular toxoplasmosis, some of the atypical findings include multifocal retinochoroiditis, absence of a chorioretinal scar, optic disc involvement, focal or widespread vasculitis, hemorrhagic vasculitis, and, most importantly, bilaterality [10,11,12]. Our patient was generally healthy, and no immunosuppressive disorders were confirmed. However, his childhood medical history was unknown. Although bilateral toxoplasmosis is observed in immunosuppressed individuals, this was not the case in our patient.

On the other hand, the unilateral character of RP lesions may suggest that the disease is caused by a somatic or germline mutation [13]. Therefore, molecular diagnosis based on blood samples may not be fully reliable. Moreover, it is currently known that RP may be caused by a mutation of one of the multiple genes (about 40) encoding proteins involved in vision or responsible for the photoreceptor structure (e.g., rhodopsin or peripherin). Additionally, more advanced genetic testing is thus indicated to solve the diagnostic dilemmas in similar complex cases. To our knowledge, only one case of a patient with coexisting RP and congenital toxoplasmosis has been reported before [14].

## 4. Conclusions

Our case illustrates challenges in the diagnosis of an atypical form of ocular toxoplasmosis leading to pigmentary retinopathy that can mimic unilateral RP. To confirm the diagnosis, the patient’s medical history should be carefully analyzed along with a fundus examination, ocular imaging, laboratory tests, and, in severe cases, genetic testing. Based on all these examinations, our patient was diagnosed with ocular toxoplasmosis with secondary pigmentary retinopathy most probably due to inflammation. However, extended molecular testing would be needed to establish a definitive diagnosis.

## Figures and Tables

**Figure 1 medicina-57-00892-f001:**
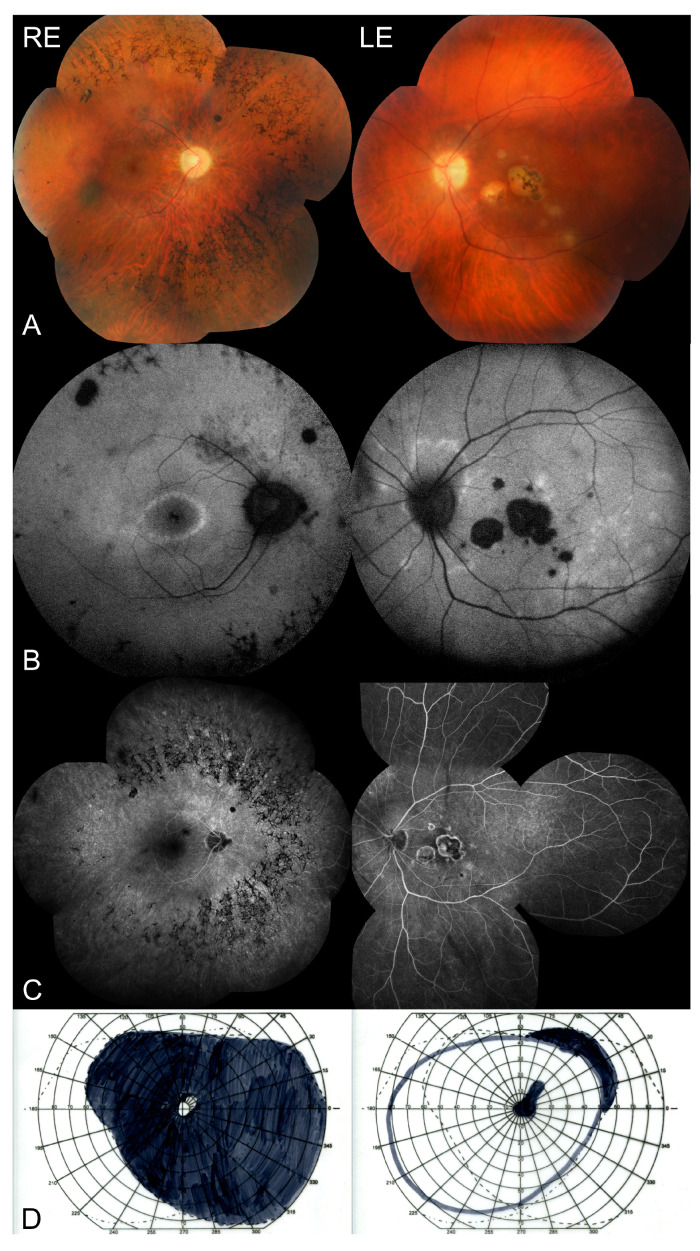
(**A**) Color fundus photography showing bone spicule-shaped pigment deposits, a slightly pale optic disc, arteriole constriction, and two small hyperpigmented chorioretinal scars in the RE, as well as an active inflammatory lesion at the border of the pre-existing chorioretinal scar in the macula in the LE. (**B**) Fundus autofluorescence showing hypoautofluorescence due to bone spicule-shaped pigment deposits and two small oval scars in the RE; central hypoautofluorescence with satellite spots in the macula of the LE. (**C**) Fluorescein angiography showing patchy hypofluorescence due to bone spicule-shaped pigment deposits in the periphery of the RE, as well as hypofluorescence in the central part of the macula in the left eye with hyperfluorescent satellite spots due to active inflammation during toxoplasmosis (LE). (**D**) Kinetic perimetry showing a concentric narrowing of the visual field in the RE and a central scotoma in the LE.

**Figure 2 medicina-57-00892-f002:**
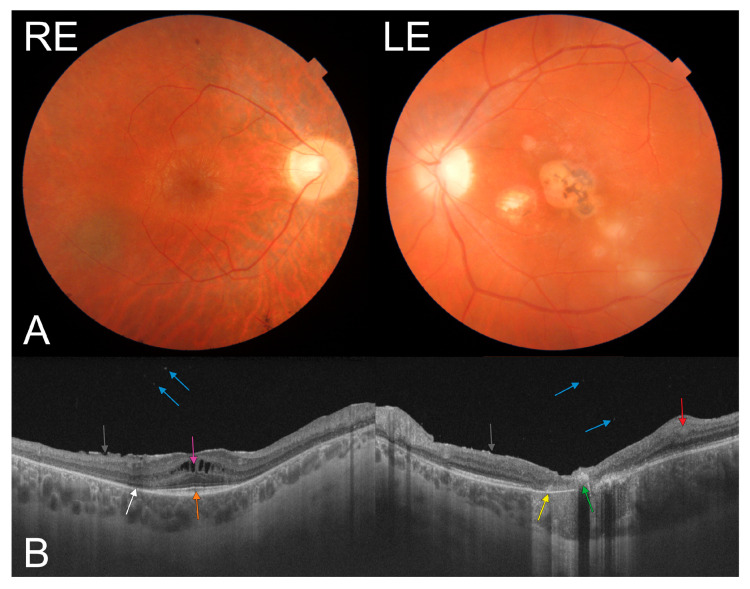
(**A**) Color fundus photography of the posterior pole showing an epiretinal membrane in the macula in the right eye (RE), as well as active inflammatory lesions at the border of the pre-existing chorioretinal scar in the macula of the left eye (LE). (**B**) Swept-source optical coherence tomography showing cystoid macular edema (pink arrow) with an epiretinal membrane (black arrow), a normal RPE and foveal photoreceptors (orange arrow) but no photoreceptor-pigment epithelium complex (white arrow) in the rest of the retina, single hyperreflective satellite spots in the vitreous of the RE (blue arrow), and atrophy of the outer retinal layers in the fovea (yellow arrow), with a hyperreflective area corresponding to pigment deposits (green arrow), and retinal thickening and disorganization of the retinal structure (red arrow) with markedly thickened choroid corresponding to an active inflammatory lesion with hyperreflective satellite spots in the vitreous (blue arrow) in the LE.

## Data Availability

Not applicable.

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
