# Peer review of "Ocular Toxoplasmosis Associated with Unilateral Pigmentary Retinopathy That May Mimic Retinitis Pigmentosa: Diagnostic Dilemmas"

_medicina, 2021, doi:10.3390/medicina57090892_

Round 1

Reviewer 1 Report

The authors describe a case of coexisting pigmentary retinopathy and ocular toxoplasmosis. 

1. If the authors consider that the findings in this patient's right eye are not unilateral RP but an atypical presentation of toxoplasmosis, what is the mechanism by which it appears? It is simply referring to inflammation, but a more detailed hypothesis or additional explanation is needed. 

2. There is a case report in which unilateral RP was caused by germline mutation of the RP1 gene (heterozygous nonsense mutation p.R677X). How about the detailed results of genetic testing of this patient?
- Mukhopadhyay R, Holder GE, Moore AT, Webster AR. Unilateral retinitis pigmentosa occurring in an individual with a germline mutation in the RP1 gene. Arch Ophthalmol. 2011 Jul;129(7):954-6.

3. page 4, line 95 "~ have been reported.5"
It should be modified in reference form. 

Author Response

We would like to thank the editors and reviewers for the effort taken to review our manuscript. We revised the paper according to all comments. We hope that the revised version satisfies all concerns and you will find it worth publishing.

Reviewer 2 Report

This is a rare case in the ophthalmic filed and qualified to be published. The experience from the authors could be shared to all the readers in the world.

However, at Line 95, we found the "5" .  Is it a mistake  ?  Please delete it.          

Author Response

(The authors gave the same response as above.)

Reviewer 3 Report

the article is interesting, but I have some doubts about the final diagnosis. Why according to the authors does the left eye have no sign of RP? What hypothesis did they formulate? I am not sure of the final diagnosis, a more in-depth genetic analysis should be made considering the molot zvocator aspect of RP in the right eye. 

Author Response

(The authors gave the same response as above.)
